# GENERATING IMAGES FROM SOUNDS USING MULTIMODAL FEATURES AND GANS

## ABSTRACT

Although generative adversarial networks (GANs) have enabled us to convert images from one domain to another similar one, converting between different sensory modalities, such as images and sounds, has been difficult. This study aims to propose a network that reconstructs images from sounds. First, video data with both images and sounds are labeled with pre-trained classifiers. Second, image and sound features are extracted from the data using pre-trained classifiers. Third, multimodal layers are introduced to extract features that are common to both the images and sounds. These layers are trained to extract similar features regardless of the input modality, such as images only, sounds only, and both images and sounds. Once the multimodal layers have been trained, features are extracted from input sounds and converted into image features using a feature-to-feature GAN. Finally, the generated image features are used to reconstruct images. Experimental results show that this method can successfully convert from the sound domain into the image domain. When we applied a pre-trained classifier to both the generated and original images, 31.9% of the examples had at least one of their top 10 labels in common, suggesting reasonably good image generation. Our results suggest that common representations can be learned for different modalities, and that proposed method can be applied not only to sound-to-image conversion but also to other conversions, such as from images to sounds.

## 1 INTRODUCTION

Generative adversarial networks (GANs), which learn through competition between generator and discriminator networks, can generate artificial but realistic data by learning the target domain's probability distribution directly (Goodfellow et al. (2014); Radford et al. (2015)). While early GAN models were trained to generate data from input consisting of latent vectors, which were assumed to obey Gaussian distributions, recent developments have enabled them to learn how to convert from one domain to another, with conditional domain data as input (Mirza & Osindero (2014)).

Such conditional GANs have proved to be broadly applicable to learning mapping functions involving conditional information such as in image-to-image translation tasks. Typical examples include converting a monochrome photograph to a colored one or filling in objects based on an edge map (Isola et al. (2016)). In addition, more advanced work has demonstrated that they can convert back and forth between two domains based on unpaired data, i.e., without needing pairs of elements, one from each domain. For example, they were able to learn a domain mapping function to convert photographs to van-Gogh-style paintings, even without having pairs of paintings and corresponding photographs to work from (Zhu et al. (2017)). However, although conditional GANs are very good at converting between two domains, more general models that can convert between different modalities, such as sound-to-image or image-to-sound, have yet to be realized, partly due to information and dimension mismatches.

There are two main issues when attempting to develop such cross-modality conversions (e.g., from sounds to images): lack of an appropriate dataset, and mismatched dimensions for the two modalities. To mitigate these issues, we first labeled the sounds and images in an unlabeled dataset separately using pre-trained classifiers and selected the data that had been given the same label in both the sound and image sets. Then, we used a pre-trained classifier to extract features from these sound/image pairs.

To realize mutual conversion across different domains with different dimensions, we have developed two unique techniques. First, we introduce multimodal layers that unify image and sound features into a set of multimodal features. Second, we introduce a feature-to-feature GAN that converts multimodal features derived from input sounds to image features, which are then converted into images by a feature-to-image GAN (Figure 1). In this way, we have been somewhat successful in reconstructing realistic images corresponding to the input sounds.

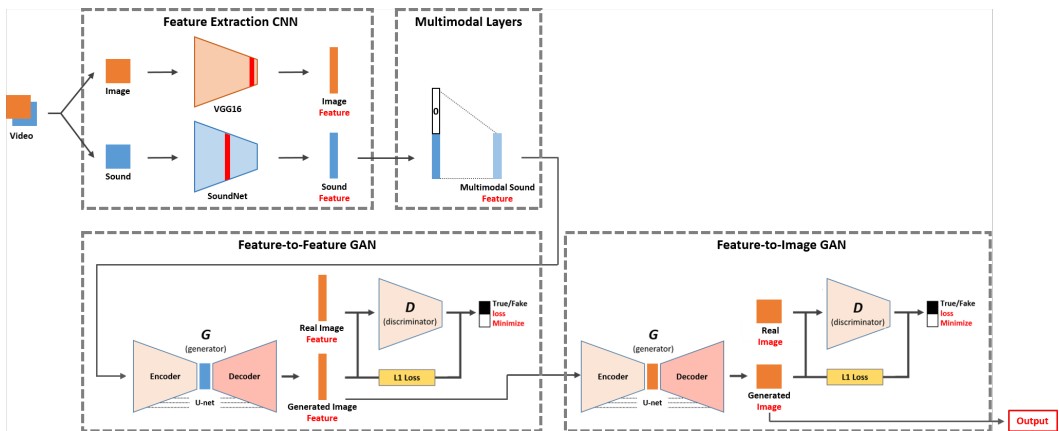

Figure 1: Model overview. First, multimodal features are created from sound features extracted from the input sounds by SoundNet. Then, two GANs are used to generate images from these multimodal features by first converting the multimodal features into image features, then converting these image features into the output images.

## 2 RELATED WORK

Deep autoencoders have achieved improved image classification performance for data with additional sound input by learning cross-modal representations (Ngiam et al. (2011)). Multimodal features generated by this bimodal deep autoencoder can also provide significant information for noisy speech classification tasks. These results indicate that such multimodal features can capture meaningful information about both sounds and images. However, the learned representations were only partly invariant to the input modality, suggesting that the shared representations could be improved.

In some cases, exploiting multiple modalities has also improved sound classification performance. For example, SoundNet's sound classification performance was dramatically improved by learning sound representations acquired via multimodal learning (Aytar et al. (2016)). SoundNet can extract common high-level representations for images and sounds by matching sound and image features. Given that SoundNet's features include image feature information, it may be possible to reconstruct images from them.

Indeed, a proposed method of converting between images and sounds is based on learning to find relationships between sounds and images in data that includes material properties and physical interactions, such as hammering (Owens et al. (2015)). By incorporating changes due to object interactions in a convolutional neural network-long short-term memory (CNN-LSTM) structure, the model could predict the corresponding waveforms. However, this model can only be applied in a restricted, highly-regular domain, whereas we are aiming to convert between a much wider range of images and sounds.

One notable study was able to successfully convert between images of musical instruments or poses and corresponding sounds (Chen et al. (2017)). Although this is quite relevant to our work, their method is limited to converting between specific sound and image types that are commonly paired in the source and target domains, and is not applicable to general video data, which include many irregularities and mismatches.

## 3 PREPARING THE TRAINING DATASET

We began by deriving a suitable subset of the SoundNet dataset (Aytar et al. (2016)). This includes more than two million unlabeled image/sound pairs, where each pair is taken from one of the videos of people's daily lives posted to flickr. We decided to use this vast number of image/sound pairs to train a model to convert sounds into images. However, one problem with this dataset is that even though the sound and image in each pair were extracted from the same video, they do not necessarily contain the same features. For example, a video of the beach may include the cries of gulls that cannot be seen in the image. Because data with this type of image/sound mismatch is unsuitable for use in training, we decided to refine the dataset.

First, we labeled all the images and sounds using pre-trained image (Inception-resnet-v2) (Szegedy et al. (2016)) and sound (SoundNet) (Aytar et al. (2016)) classifiers, respectively. Then, we selected only instances where the image and sound had at least one of their top 3 labels in common. In order to include equal numbers of data points in each class, we selected 52 classes with 2,000 instances each (for a total of 104,000 matching pairs) from more than two million videos. Figure 2 shows some examples of the images selected, which appear to have characteristics typical of their classes. We used 90% of the data for training and the remaining 10% for evaluation. The sounds were converted to single-channel MP3 audio at a sampling rate of 22 kHz, scaled to range between $-256$ and $256$.

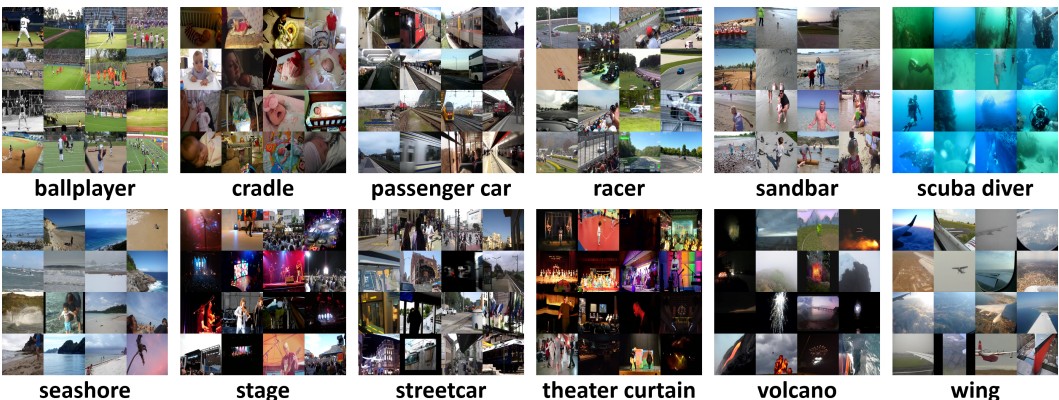

**ballplayer     cradle     passenger car     racer     sandbar     scuba diver**

**seashore     stage     streetcar     theater curtain     volcano     wing**

Figure 2: Labeled image/sound pair dataset. We refined the dataset by selecting only pairs where the image and sound had at least one of their top 3 labels in common.

## 4 FEATURE EXTRACTION

We first extracted image and sound features using pre-trained CNN models designed for image and sound classification, respectively. To extract image features, we used VGG-16 (Simonyan & Zisserman (2014)), as it provides high classification performance with a simple structure. Although state-of-the-art CNN models with deep and wide structures can provide slightly better performance, they would not have been suitable for our study because their structures are too complicated to extract feature vectors. In general, the earlier layers contain a lot of information, but they also contain noise, such as position information that is uncorrelated with features representing object-related concepts. In contrast, the deeper layers include less positional information, focusing more on conceptual features. In this study, we extracted the image features from VGG-16's fc6 layer, as this has little spatial information but substantial object-related class information, and is also easy to use for training, because its vector is significantly smaller (4096) than that of the last convolution layer (25088).

To extract sound features, we used SoundNet (Aytar et al. (2016)), which was designed to learn image representations from sounds. We extracted the sound features from SoundNet's pool5 layer, because this layer's features yield the best sound classification performance, indicating that it includes highly-refined sound feature information. Whereas deeper layers represent information that

is more focused on the corresponding image, the pool5 layer retains more sound information and is thus more suitable for our purposes.

# 5 CONVERTING SOUNDS INTO IMAGES

## 5.1 CONDITIONAL GANS

In order to convert sounds into images, we need to remove noise and extract features that are common to both the sound and image. However, this also results in less information being available, causing serious overfitting when attempting to recover image from it. To solve this problem, GANs, which can learn the target domain's probabilistic distribution, can be useful. In this study, we used a conditional GAN. Its generator network has a U-Net structure, where each encoder layer is directly connected to each decoder layer. This has the advantage that it can use conditional input in the conversion process. In general, encoder-decoder structures are better at reducing noise and distilling information about meaningful representations. At the same time, however, since the information is reduced using bottleneck layers, severe overfitting can occur when reconstructing images from a small amount of information. The U-Net structure deals with this information reduction issue by transferring the information to the back side while retaining the advantage of extracting meaningful features.

When utilizing a conditional GAN to convert between different domains (such as sound features to images), there is a risk that it will learn a simple one-to-one conversion. To resolve this issue, we concatenate multiple GANs, using a strategy similar to that of stackGAN (Zhang et al. (2017)), which facilitates learning by subdividing the steps of the mapping route. Instead, of performing direct (e.g., pixel-to-pixel) spatial conversion, we first convert the features from one domain to the other using the first conditional GAN. Then, we convert the generated features into an image using a second conditional GAN.

## 5.2 CONVERTING IMAGE FEATURES INTO IMAGES

Our ultimate goal is to convert refined sound features (with less information) into images (including a larger amount of information). As described above, the idea is for a conditional GAN to recover more information from the reduced information by learning the target domain's probabilistic distribution. To confirm that the conditional GAN was indeed suitable for this problem, we tested whether it could convert image features into images. Image feature vectors (4096 dimensions) were treated as two-dimensional $64 \times 64$ images, then up-sampled to $128 \times 128$ in order to obtain the ensemble effect. This confirmed that the images could be restored from the image features using the conditional GAN (Figure 3). This ability to convert image features into images indicates that, if sound features can be converted into image features, images can then be derived from them. We therefore decided to use this strategy.

## 5.3 MULTIMODAL FEATURES

A significant issue when using a conditional GAN to convert between two different feature domains is that the discriminator may converge too rapidly while the generator does not learn sufficiently. This can happen when two different domains are irregular with little matching information, which makes it difficult for the generator to learn a suitable mapping between them, while the discriminator can easily distinguish real from generated data. In addition, this mismatch issue can result in one-to-one conversion, rather than learning features that are common to both the image and sound. To avoid this, we introduce multimodal layers to extract such common features.

Our multimodal layers consist of input, hidden, and output layers. To train the network, we trained the hidden layer vectors on three different types of inputs: images only (I), sounds only (S), and both images and sounds (B) (Figure 4). When the distributions of the hidden layer outputs for I and S converge, the output value for B is twice that for I or S. We therefore used half the hidden layer output value for B, together with the output values for I and S, for training.

To train the multimodal layers on these three different inputs (I, S, and B), we considered three approaches: hidden layer L1 loss, hidden layer cross-entropy, and output layer cross-entropy. Al-

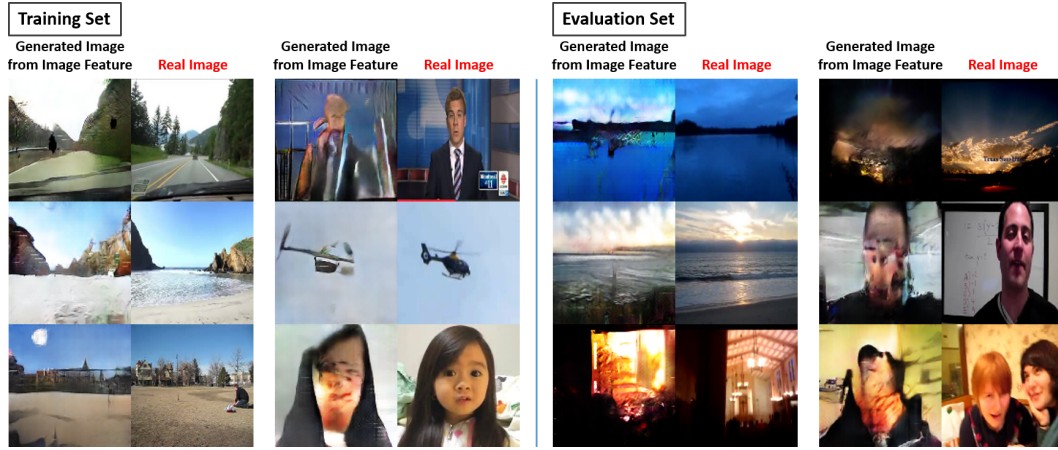

Figure 3: Images generated from image features. Here the generated images retained the original images' characteristics, such as their spatial and color distributions and the objects present, despite being restored from reduced information.

though the L1 loss is the simplest way to train such vectors, it depends on the magnitude of the vector's value. In order to quantify the loss independently of this, we included the cross-entropies of the hidden layer outputs for I and S. The cross-entropy loss was set not only after but also before the activation function. The reason for including the loss before the activation function is to extract similar information before the information reduction and to transmit this similar information after the activation function. Minimizing the L1 loss for the hidden layer between, say, I and S can result in zero vectors in the hidden layer for both I and S. To avoid this problem, we used the cross-entropy loss for the classification labels in the output layer. We minimized these three losses using the Adam optimizer (Kingma & Ba (2014)).

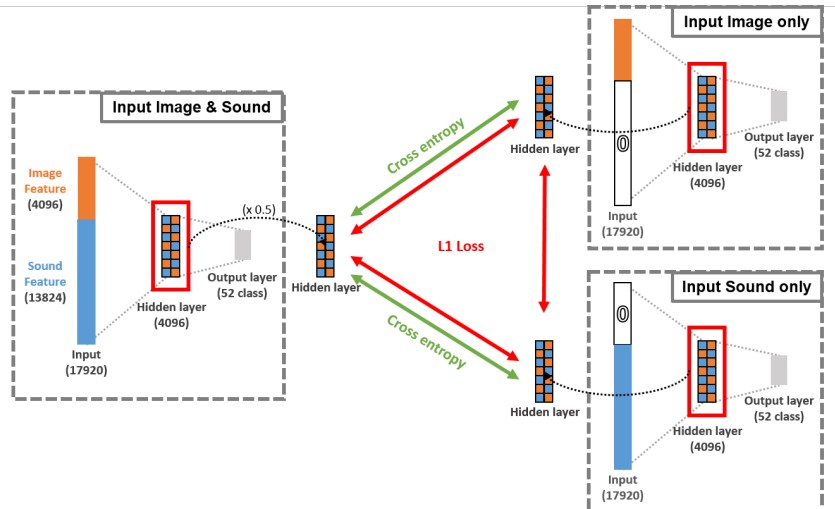

Figure 4: Extracting multimodal features by training the multimodal layers. This figure shows how the proposed multimodal layer (used to reduce the information mismatch between the two modalities) is structured and trained, and also how the multimodal features are extracted. The multimodal features are extracted by inputting either both image and sound features, or either feature type individually.

To test whether this process generated suitable multimodal features, we compared each image with that corresponding to the sound whose multimodal features were closest (in terms of Euclidean

distance) to those of the image (Figure 5). We found that the two images mostly belonged to the same class. For the multimodal features before the activation function, we found that the class labels matched 28.2% and 16.3% of the time for the training and evaluation sets, respectively, whereas after the activation function the match rates were 47.1% and 25.3%, respectively. This shows that our multimodal learning process yielded vectors encoding information common to the image and sound, not just a one-to-one correspondence.

For sound-to-image conversion training, we decided to use the linearly-converted multimodal features from before the activation function in the hidden layer, because this had the advantage of generating sufficiently accurate common features without reducing the information available to train the GAN generator.

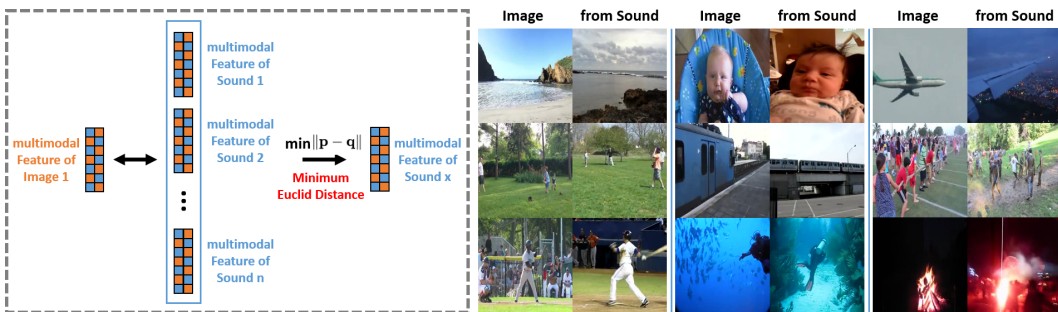

Figure 5: Visualization of multimodal features. To test whether the generated multimodal features encoded features common to the image and sound, we found the sound/image pairs whose multimodal features were closest (in terms of Euclidean distance). Here, we compare the image and the image corresponding to the sound, showing that they mostly belong to the same class.

## 5.4 CONVERTING MULTIMODAL SOUND FEATURES INTO IMAGE FEATURES

The above analysis suggests that a conditional GAN can convert features with less information into an output with more information without suffering from overfitting, due to adding suitable information about the target domain's distribution. In addition, using multimodal features benefits the GAN's generator by extracting similar information about both domains. Since we were able to convert image features into images (Section 5.2), if we can convert multimodal features derived from input sounds into image features, we can then convert these into images by using such a conditional GAN. Therefore, we first convert multimodal sound features into image features using another conditional GAN.

Multimodal sound features with 4096 dimensions were converted into two-dimensional $64 \times 64$ images, which were then up-sampled to $128 \times 128$ in order to obtain the ensemble effect. We imposed a penalty of 0.001 on the discriminator loss in order to prevent it converging too quickly during training. The up-sampled multimodal sound features were converted into image features by the first conditional GAN, and these image features were then converted into images using the second conditional GAN, trained as in Section 5.2.

Figure 6 compares some example images generated from sounds with the original images corresponding to the sounds. By introducing multimodal layers and utilizing two conditional GANs, we were able to successfully convert from the sound domain to the image domain. Although the generated images are blurred, they retain the original images' characteristic features. For example, a generated image in the "seashore coast" class has water, sky, and boundary characteristics, while an image in the "wing" class shows sky-like colors and a shape like an airplane wing. Overall, the model learned common class elements such as boundary and color accurately.

To quantify the generator's accuracy, we applied a pre-trained classifier (Inception-resnet-v2) to both the generated and original images. This assigned the 10 most probable labels, based on the images' characteristics. Figure 7 shows some example pairs of generated and original images with matching labels within the top 1, 3, 5, or 10 classes. The overall match rates were 1.65%, 8.9%, 16.04%, and 31.92%, respectively.

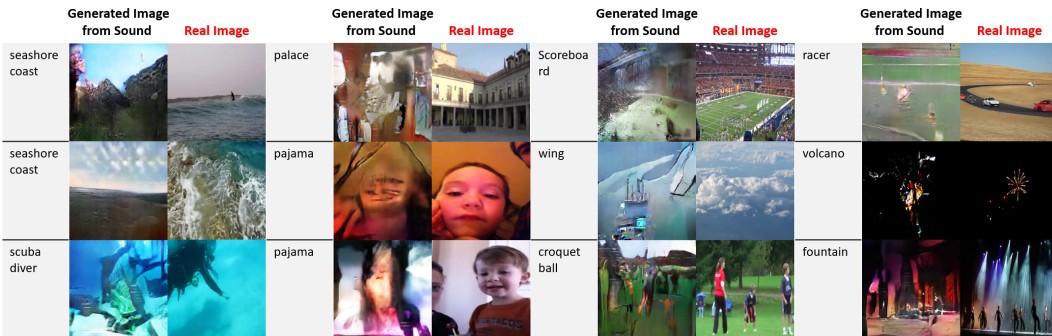

Figure 6: Images generated from sounds. Here, we show example images generated from sounds. The generated images appear to have acquired many of the characteristics of the real images, such as their spatial arrangement and color distribution.

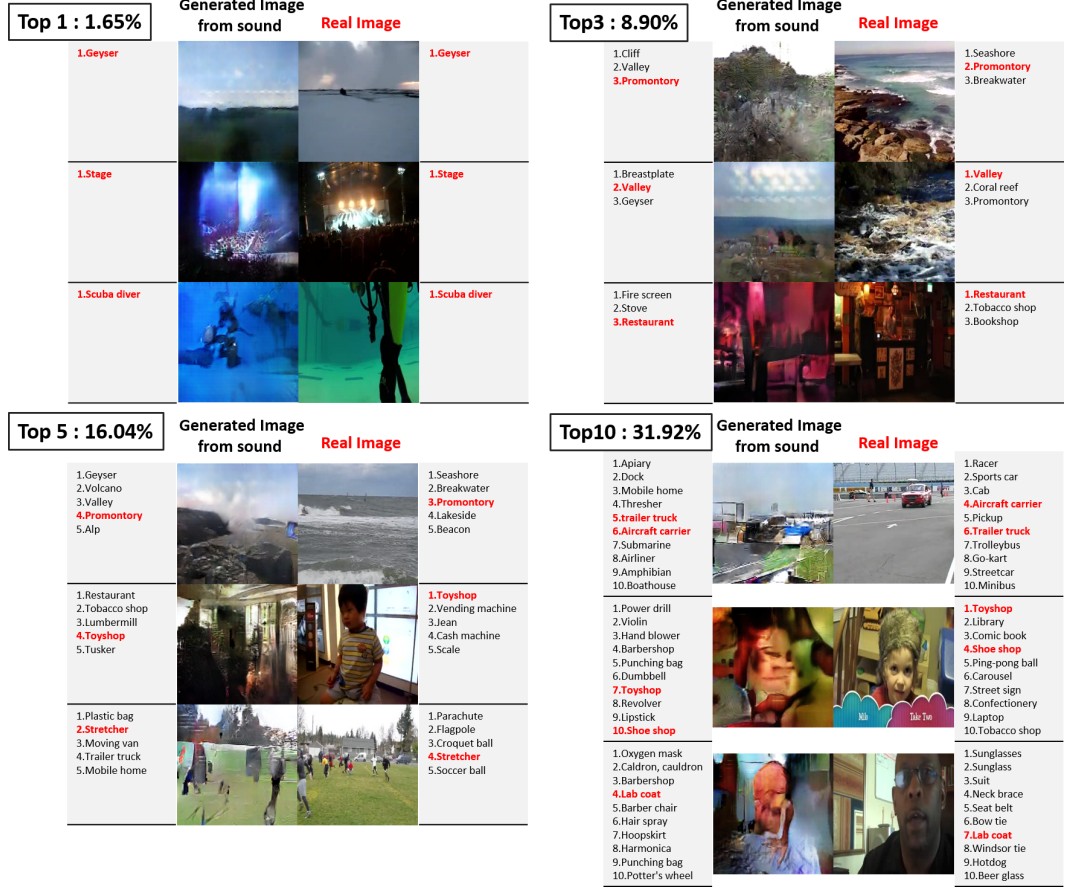

Figure 7: Model performance evaluation. Here, we show the match rates for the classes discriminated by the classifier, considering between 1 and 10 of the top classes. The overall match rates were 1.65%, 8.9%, 16.04%, and 31.92% for the top 1, 3, 5, and 10 classes, respectively.

The images where the top classes matched were often of natural scenes, such as "geyser", "valley," or "promontory", with the generated and original images being very similar. The generator learned the mapping function easily for the natural scene classes, because these scenes shared many common elements, such as sky, ground, and boundaries. Even when the images only matched within the top 10 classes, we still found that the generated and original images had some similar characteristics.

The "toyshop" class, for example, often included children, and the generated images contained children's faces or similar interior colors.

However, although the generated images often retained the original images' characteristics, amoeba-like shapes appeared in many of them and were classified as, for example, wine bottles or plastic bags, reducing the match rate. For example, in Figure 7, in the third of the top 5 match pictures, the generated image was assigned to the "plastic bag" class, despite the colors and boundaries being similar to those of the original image. A potential method to avoid the formation of amoeba-like shapes is to use a more advanced learning method such as boundary equilibrium generative adversarial networks (BEGAN) (Berthelot et al. (2017)) to further balance the learning of generators and discriminators or to stabilize learning using advanced normalization, for example the spectral normalization of GAN (Miyato et al. (2018)).

Finally, another reason for the low match rates is the pre-trained classifier's limited precision. For example, in the top 5 example discussed above, the original image was considered more likely to be a "parachute" than a "stretcher" or "soccer ball", going against human intuition. Using a better classifier might therefore improve the match rate between the generated and original images.

## 6 DISCUSSION

In this paper, we have proposed a method of converting between two modalities (images and sounds) based on multimodal features and stacked GANs. We have also shown that multimodal layers can extract similar information and generate vectors with similar distributions from both input modalities. Our method converts multimodal features derived from input sounds into image features, and subsequently into images, using conditional GANs.

The Pix2Pix model used here was originally developed to convert data between similar domains, such as from one type of image to another. Although this model's U-Net structure offers information reduction advantages similar to those of ResNet's "skip connections", the generated output is constrained by the fact that it transmits the vector's spatial information directly to the deep layer. Since vectors extracted from two different modalities generally have different information dimensionalities, these spatial information constraints may interfere with learning. Although it remains unclear how much the reduced information in this structure affects learning, it may be helpful to consider solving these problems using a simple encoder-decoder structure in future.

There are also more advanced methods of learning mapping functions between the source and target domains, such as DiscoGAN (Kim et al. (2017)). DiscoGAN can learn the mappings needed to convert data between two domains that are not in one-to-one correspondence, and can exhibit higher performance than conventional GANs when learning the mapping functions between images and sounds with similar characteristics but different dimensions. It may therefore be possible to use DiscoGAN to achieve more realistic sound-to-image and/or image-to-sound reconstruction in future.

In this study, we constructed our own dataset of sound/image pairs, including 104,000 pairs belonging to a total of 52 classes. Better reconstruction results could be obtained by simply increasing the amount of data and applying suitable normalization methods. In addition, although the multimodal layers introduced herein can yield a vector space of features common to two quantities, it is unclear whether these features are interpretable. Instead, generating a more interpretable vector space using a vast amount of data with high regularity between images and sounds may be possible. In the present study, by introducing learning to unify features between images and sounds, sounds could be converted into images. In the future, we anticipate the multimodal features introduced herein to be used for other applications such as converting sounds to images.

ACKNOWLEDGMENTS

JL was supported by the Rotary Yoneyama Memorial Foundation via a scholarship to work abroad. KA was supported by Grant-in-Aid for Scientific Research (16H05862).

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
