# OpenReview forum: "Generating Images from Sounds Using Multimodal Features and GANs"
_ICLR.cc/2019/Conference_

### Official Review · AnonReviewer2 · 2018-10-30
**A good idea, poor development and results.**

**Rating:** 4
**Confidence:** 5

**Review:**

The authors present a novel method for generating images from sounds using a two parts model composed by a fusion network, aka. multi-modal layers, for learning sound and visual features in a common semantic space, and two conditional GANs for converting sound features into visual features and those into images. To validate their approach they created an ad-hoc dataset, based on Flickr-SoundNet dataset, which contains 104K pairs of sounds and images with matching scene content. Their model was trained as two separate models, the fusion network was trained to classify both images and sounds minimizing their cross-entropy and their L1 distance, while the two conditional GANs were trained until convergence penalizing the discriminator to prevent fast convergence.

Although the idea of generating images from sounds with the aid of Generative Adversarial Networks is quite novel and interesting, the paper exhibits several problems starting with the lack of clarity explaining the purpose of the proposed method and the contributions of the work itself. Overall, the idea is good but not well developed. Introduction should present more clearly the problem and framework.

In the related work section the authors omitted some relevant recent prior works such as “Look, Listen and Learn” paper by Arandjelović and Zisserman presented on ICCV’17, “Objects that Sound” by Arandjelović and Zisserman presented on ECCV’18, “Audio-Visual Scene Analysis with Self-Supervised Multisensory Features” by Owens and Efros presented on ECCV’18, and “Jointly Discovering Visual Objects and Spoken Words from Raw Sensory Input” by Harwath et al. also presented presented on ECCV’18. These works propose different methods for aligning visual and sound features.

There are also several concerns on the validity of the results: 1) none of the results achieved by training their multi-modal layers were validated against a baseline, e.g. evaluating the quality of the learned visual features against VGG or a simple GAN instead of two stacked conditional GANs, 2) it is not clear why they learned features minimizing L1 loss + Cross-Entropy while using L2 distance to address the quality of their learned features, a simple way of doing so would be evaluating their retrieval capabilities using any standard measure from the retrieval community, e.g. the normalized discriminative cumulative gain (nDCG) or the classical mean-average precision (mAP) as proposed in “Objects that Sound”, 3) the authors assume that using a conditional GAN is suitable for generating images from visual features, but they don’t provide any quantitative results supporting this claim, they only provide a few successful qualitative results and elaborate their model from there. 4) Ablation is completely missing: it would be interesting to prove the effective contribution for i) the multi-modal fusion ii) the two-steps of image generation iii) the L_ losses for the two GANs.

There are many missing citations throughout the paper, in particular: 1) the concatenation of visual and sound features followed by a fusion network for learning features in a common semantic space was already proposed on “Look, Listen and Learn”, 2) when the authors describe their strategy for sound features extraction in section four, they never mentioned that the idea of using pool5 layer features was already introduced by SoundNet authors, and 3) in section 5.3 when they mention that using a conditional GAN to convert between two different feature domains it might be that the discriminator may converge too rapidly while the generator does not learn sufficiently.

Finally although using an ad-hoc extremely simplified dataset with pairs of images and sounds matching scene content, the complete model is able to generate images which achieve only a 8,9% matching rate for the top 3 predicted classes. Given that the dataset was created with 100% matching on the top 3 scores for sound and images, the results are definitely  poor.

---

### Official Review · AnonReviewer1 · 2018-11-05
**Generating Images from Sounds Using Multimodal Features and GANs**

**Rating:** 4
**Confidence:** 4

**Review:**

Summary:

This paper addresses the problem of generating images from sound. The general idea is to use conditional GANs. In particular, two stacked conditional autoencoder GANs, where the autoencoders have a U-Net architecture. First, sound features are mapped into multimodal features that contain image feature/class information. Such multimodal features condition the generation of image features with an initial GAN, and the image features condition the generation of the output image with a second GAN. Although the problem itself is rather difficult, the solution is almost entirely based on previous work. The most novel part of the paper is the learning of the multimodal features. The final results are not very compelling, the literature review is very limited. There is a very high-level description of the approach with very few details, which leave the reader with a lot of unanswered questions. There is no attempt to compare with previous work, the architecture is not studied in depth with an ablation study, or compared with more interesting baselines, other than verifying that images can be generated from features (something not very surprising, given StackGUN!). Probably the more important verification is that the multimodal feature can embody some class information. Overall it seems to be a limited contribution.

Comments on quality, clarity, originality and significance:

This paper provides a high-level description for an approach to a problem that is relatively difficult to address. The paper is not very well motivated, and therefore lacks clarity and leaves the reader with a lot of unanswered questions. The literature review is limited as it is the set of results and comparison with previous works.

The paper addresses an important problem; however, I feel the work is not very significant because it does not reveal new techniques, nor produces compelling results, nor performs a deep analysis.

I think the problem is interesting, but a deeper analysis is needed to lift this contribution up to a significant one.

Below is a summary of some of the additional questions gathered while reading the paper:

Why there is the need for the multimodal features at all? Why can’t the sound be converted into class labels and then StackGAN can generate images?

Why do you need a feature-to-feature GAN? Why not generating images directly from the generated multimodal features? Motivations are not provided clearly.

Unclear architecture from Figure 1. The Feature-to-Feature GAN and the Feature-to-Image GAN have the same architecture? What does the encoder and decoder do? How are they organized?

Looks like every piece is trained alone, no end-to-end learning, right? Please clarify that point.

No attempt to compare with other approaches has been made. Also, no effort to formulate a baseline model. What would happen if one were to use solely the features generated by SoundNet?

Would you be able to compare your multimodal features with those generated by Ngiam et al. (2011), for instance?

Section 3 refers to a 90/10 training/evaluation split but then it is unclear in what experiments that exact split is used.

No description on hyperparameters.

No complexity, no architecture details, (also no equations that could provide more details).

It should be clarified what it means one-to-one conversion. It is brought up in several points in the paper, but it is never clear what it means and therefore how it relates to what the Author intends to stress.

It is unclear why by performing first a feature-to-feature mapping and later a feature-to-image mapping the one-to-one conversion problem should be addressed. In StackGAN the problem addressed is the resolution increase. The problem addressed by this paper is unclear.

Unclear what is the “ensemble effect”, and what is the motivation for upsampling a feature vector in two dimensions.

What image features were used to generate the images in Figure 3? Which architecture was used and how was it trained? Where these the same multimodal features used in the full architecture?

The paragraph motivating the need for multimodal features is unclear.

How are the three type of losses weighted for learning the multimodal layers? No discussion provided on that.

Unclear why a multimodal vector should be upsampled in 2 dimensions.

The training procedure and loss for training the feature-to-feature conditional GAN is not explained.

Despite the difficulty of the problem, the generated images do not look compelling.

16 references do not seem enough by todays’ high-quality standard conferences.

---

### Official Review · AnonReviewer3 · 2018-11-08
**I think the problem is ill-posed; the image generations from image features are not great; baselines from class labels would have worked beter; lacks motivation.**

**Rating:** 3
**Confidence:** 4

**Review:**

PROS:
* The paper was well-written and explained the method and the experiments well

CONS:
* The problem seems ill-posed to me. Sound is temporal and the problem should probably be sound-to-video conversion not sound-to-image.
* A link to generated images from sounds where one could actually evaluate the generations would be useful. Currently the only way to evaluate the results is via labels.
* Similarly, a baseline where images are generated given the classification labels of the sounds would probably produce better looking images. Such baseline is not provided, and it is not clear to me what a multi-modal feature extraction is providing on top of this.  For example, in the case of StackGAN, the GAN that was converting text to images, the text was describing something about the image that one could quantify in the resulting generation (eg a blue bird as opposed to a yellow one). Here such an advantage is not clear and if there is one, it should be clearly stated and discussed.
* The results in Fig. 3 seem particularly poor and on par with current GAN generations. I think this part of the model should be improved before attempting to improve the rest.
* In Figures 6 and 7, it is not clear what we are expected to see. Also, the labels do not correspond to the real images in many of hte cases (eg pajama, wing, volcano etc).


Finally in the discussion, DiscoGAN is mentioned as something to look into for future work. I should note that DiscoGAN is converting samples between domains of the same modality (vision), in the context of domain adaptation, similarly to other works.

---

### Meta-Review · Area_Chair1 · 2018-12-15

**Confidence:** 5
**Recommendation:** Reject

**Metareview:**

The work presents a new way to generate images from sounds. The reviewers found the problem ill-defined, the method not well-motivated and the results not compelling. There are a number of missing references and things to compare to, which the authors should change in a follow-up.